# Cisplatin-induced DNA double-strand breaks promote meiotic chromosome synapsis in PRDM9-controlled mouse hybrid sterility

Liu Wang[1][†][‡], Barbora Valiskova[1,2][†], Jiri Forejt[1]*

[1]BIOCEV Division, Institute of Molecular Genetics, Czech Academy of Sciences, Vestec, Czech Republic; [2]Faculty of Science, Charles University, Prague, Czech Republic

*For correspondence:
jforejt@img.cas.cz

[†]These authors contributed equally to this work

Present address: [‡]Department of Genetics and Genome Sciences, School of Medicine, Case Western Reserve University, Cleveland, United States

Competing interests: The authors declare that no competing interests exist.

**Abstract** PR domain containing 9 (*Prdm9*) is specifying hotspots of meiotic recombination but in hybrids between two mouse subspecies *Prdm9* controls failure of meiotic chromosome synapsis and hybrid male sterility. We have previously reported that *Prdm9*-controlled asynapsis and meiotic arrest are conditioned by the inter-subspecific heterozygosity of the hybrid genome and we presumed that the insufficient number of properly repaired PRDM9-dependent DNA double-strand breaks (DSBs) causes asynapsis of chromosomes and meiotic arrest (*Gregorova et al., 2018*). We now extend the evidence for the lack of properly processed DSBs by improving meiotic chromosome synapsis with exogenous DSBs. A single injection of chemotherapeutic drug cisplatin increased frequency of RPA and DMC1 foci at the zygotene stage of sterile hybrids, enhanced homolog recognition and increased the proportion of spermatocytes with fully synapsed homologs at pachytene. The results bring a new evidence for a DSB-dependent mechanism of synapsis failure and infertility of intersubspecific hybrids.
DOI: https://doi.org/10.7554/eLife.42511.001

## Introduction

Proper synapsis of homologous chromosomes is an important meiotic checkpoint preventing germ-line transfer of harmful genic and chromosomal mutations to the next generations (*Schimenti, 2005*; *Zickler and Kleckner, 2015*; *Rinaldi et al., 2017*). Synapsis of homologous chromosomes is initiated at the leptotene stage of the first meiotic prophase by induction of developmentally programmed, SPO11-induced DNA double-strand breaks (DSBs) (*Keeney et al., 1997*; *Keeney et al., 1999*; *Romanienko and Camerini-Otero, 2000*). After 5' to 3' resection of each end of DSB, replication protein A (RPA) binds the 3' overhang to save it from degradation, later being displaced by RAD51 and DMC1 recombinases (*Inagaki et al., 2010*) but see (*Moens et al., 2007*; *Chan et al., 2018*). The resulting nucleoprotein filament is engaged in homology search in the process leading to DSBs repair by homologous recombination and to synapsis of homologous chromosomes. The SPO11-induced DSBs are nonrandomly clustered into narrow 1–2 kilobase-pair intervals called recombination hotspots and their localization is predetermined by the PRDM9 binding to specific motifs inside these intervals and PRDM9-driven induced trimethylation of histone H3 at lysine 4 and lysine 36 on adjacent nucleosomes (*Baudat et al., 2010*; *Myers et al., 2010*; *Parvanov et al., 2010*; *Eram et al., 2014*; *Lange et al., 2016*; *Powers et al., 2016*, for recent reviews see *Grey et al., 2018* and *Paigen and Petkov, 2018*).

The *Prdm9* gene, besides determining position of the recombination hotspots, acts as the major hybrid sterility gene in certain hybrids between house mouse subspecies of *Mus m. musculus* (mouse

strain PWD) and *Mus m. domesticus* (mouse strain C57BL/6, hereafter B6) (*Mihola et al., 2009*; *Dzur-Gejdosova et al., 2012*; *Forejt et al., 2012*; *Bhattacharyya et al., 2013*; *Bhattacharyya et al., 2014*). Disrupted synapsis of homologous chromosomes and dysregulation of meiotic sex chromosome inactivation are two major cellular phenotypes controlled by the *Prdm9* gene in sterile (PWD x B6)F1 (hereafter PBF1) hybrids (*Forejt and Iványi, 1974*; *Mihola et al., 2009*; *Bhattacharyya et al., 2014*; *Gregorova et al., 2018*).

When the mouse *Prdm9* gene was 'humanized' by substitution of the C2H2 zinc-finger (ZnF) DNA-binding domain for its human ortholog, the humanized PBF1-*Prdm9*$^{Hu/PWD}$ meiocytes regained normal meiotic pairing and hybrid males became fertile. This unexpected finding provided direct evidence for the role of PRDM9 ZnF array in the control of hybrid sterility (*Davies et al., 2016*). The asynapsis and male sterility were proposed to be mainly a consequence of the evolutionary erosion of PRDM9 binding sites (*Figure 1*). Because the heterozygous allelic sites with lower PRDM9 binding affinity are used preferentially as a template for DSB repair in gene-conversion events, the sites with higher binding affinity mutate much faster than the rest of the genome. As a result, the majority of the PRDM9$^{PWD}$-determined hotspots in PBF1 sterile males are found on B6 homologs and vice versa. Such hotspot asymmetry can result in a delay or inability to repair DSBs using homologous chromosome as a template, thus preventing successful pairing and synapsis of homologs (*Davies et al., 2016*).

We recently rescued synapsis of homologous chromosomes in meiosis of PBF1 sterile hybrids by elimination of PRDM9 hotspot asymmetry in random chromosomal intervals (≥27 Mb), with paternal and maternal copies originating from the same PWD subspecies (*Gregorova et al., 2018*). When synapsis of the four chromosomes most strongly affected by asynapsis in the sterile hybrids was restored in this way, male fertility was regained (*Gregorova et al., 2018*). To further test the idea

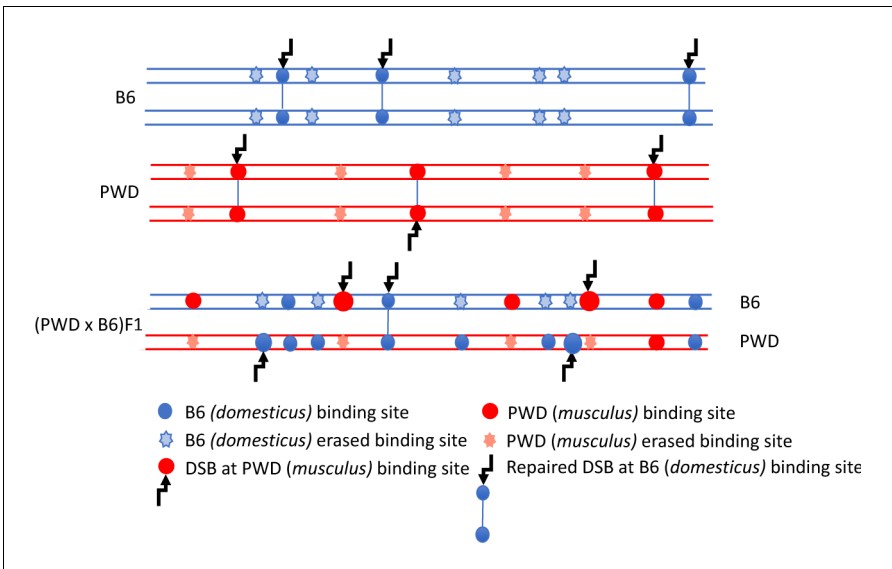

**Figure 1.** DSB asymmetry model based on historical erosion of PRDM9 binding sites. A simplified scheme of a pair of homologous chromosomes in PWD (*Mus m. musculus*) and B6 (*Mus m. domesticus*) mice and sterile (PWD x B6) intersubspecific male F1 hybrids. Eroded PRDM9$^{B6}$ binding sites are not recognized or hardly recognized by the PRDM9$^{B6}$ zinc-finger array in B6 meiosis, but the same sites were saved from erosion during the evolution of the other subspecies. Thus, in (PWD x B6)F1 hybrids PRDM9$^{B6}$ often binds to the sites on PWD chromosome that are erased on B6 homolog and, vice versa, PRDM9$^{PWD}$ more often binds to the sites on B6 homolog, eroded in PWD. The proportion of such asymmetric sites exceeds 70% of all DSBs in (PWD x B6)F1 hybrid meiosis (*Davies et al., 2016*) and interferes with chromosome synapsis and meiotic progression. The higher activity of these asymmetric hotspots estimated by DMC1-ChIP-seq is explained by a delay or failure of DSB repair.
DOI: https://doi.org/10.7554/eLife.42511.002

that the failure of proper meiotic synapsis in hybrid males is due to an insufficient number of timely repaired 'symmetric' DSBs and to evaluate the unlikely possibility that the rescue was caused by multiple recessive genetic factors of PWD origin, we increased the number of DSBs per cell by inflicting random exogenous DSBs to meiotic cells. We report that the exogenous DSBs generated by chemotherapeutic drug cisplatin (*Basu and Krishnamurthy, 2010*) enhanced meiotic synapsis of homologous chromosomes in sterile mouse inter-subspecies hybrids, thus bringing independent evidence on the mechanism of meiotic chromosome asynapsis (*Gregorova et al., 2018*) and supporting the 'asymmetry' hypothesis (*Davies et al., 2016*).

## Results and discussion

Cisplatin (cis-platinum diamminedichloride, hereafter cisPt) is known to create DNA inter-strand (ICL) and intra-strand cross-links. In replicating yeast and mammalian somatic cells removal of ICLs results in DNA DSBs, which can be repaired by the nonhomologous end joining or by homologous recombination, the latter being favored in the germline (*Lawrence et al., 2016*). Removal of the cisPt-DNA adducts without creating DSBs was reported in quiescent somatic cells (*Frankenberg-Schwager et al., 2005*). In the mouse, cisPt was reported to increase meiotic crossing-over (*Hanneman et al., 1997*). Moreover, significant improvement of meiotic chromosome synapsis was observed in SPO11-/- meiocytes treated with cisPt or X rays, indicating that exogenous DSBs can at least partially substitute the role of the SPO11-induced DSBs in pairing of homologous chromosomes (*Romanienko and Camerini-Otero, 2000*; *Carofiglio et al., 2018*).

To assess an effect of exogenous DSBs on meiotic pairing in sterile hybrids we treated the adult (4–8 weeks) PBF1 hybrid males with cisPt and with the 5-ethynyl-2'-deoxyuridine (EdU), a nucleoside analog of thymidine (*Salic and Mitchison, 2008*) to distinguish the spermatogenic cells replicating their DNA at the moment of cisPt injection (*Figure 2A*). The males received a single i.p. injection of cisPt at a dose of 1, 5 or 10 mg/kg body weight together with 50 mg/kg of EdU. Based on the published estimates of duration of the meiotic S-phase (20 hr), leptotene (24–48 hr), zygotene (24–32 hr) and pachytene (160 hr) stages of the first meiotic prophase (*OAKBERG, 1956*; *Oud et al., 1979*; *Goetz et al., 1984*) the males were sacrificed 40 hr after cisPt and EdU injection to quantify the DSBs at the first meiotic prophase, or after 8 days to monitor the chromosome synapsis at the pachytene stage.

### CisPt induced DSBs in early meiotic prophase of sterile male hybrids

Forty hours after cisPt and EdU injection, 84.1 ± 3.3% (mean ±SE) of leptonemas and 49.3 ± 2.2% of zygonemas were EdU-positive, thus being at the S-phase at the time of injection or shortly after that. The EdU-negative leptonemas (15.9%) most likely started their S-phase 20 hr or more after EdU injection, after assumed depletion of free EdU (*Figure 2B*, *Figure 2—source data 1*), while EdU-negative zygonemas finished their DNA replication before CisPt and EdU injection. All pachynemas finished DNA replication before the treatment and were EdU-negative (*Figure 2B*). The occurrence of EdU-positive cells fitted better with the shorter reported estimates of the duration of leptotene and zygotene stages.

Since the RPA protein was reported to bind ssDNA soon after resection of DSBs in mitotic and meiotic cells (*Ribeiro et al., 2016*; *Pacheco et al., 2018*), we used the RPA foci as an early cytological marker of DSBs (*Figure 3A*). We found out that in spite of the large variation in the number of RPA foci in individual leptonemas and zygonemas (see also *Kauppi et al., 2013*), cells treated with 10 mg/kg of cisPt showed significant increase of RPA foci in leptotene and zygotene stages (*Figure 3B*, *Figure 3—source data 1*). The leptotene median number of 162 RPA foci per cell in control males increased to 229 foci in males treated with 10 mg/kg of cisPt (p=0.0313 Mann-Whitney U test). Control zygotene median of 194 RPA foci increased to 210.5 foci after treatment with 10 mg/kg of cisPt (p=0.0483). The numbers of RPA foci decline at the pachytene stage of meiotic prophase in fertile male mice (*Li et al., 2007*; *Inagaki et al., 2010*) but persist in high numbers in sterile untreated hybrids (median 119 RPA foci per pachynema). CisPt did not affect frequency of RPA foci in pachynemas 40 hr after injection (*Figure 3B*). To evaluate the impact of RPA foci on pachynemas and because beside asynapsis, the unrepaired DSBs are known to induce apoptosis, we compared the frequency of RPA foci in early, mid and late pachynemas of control (0 mg/kg of cisPt) sterile PBF1 males with fertile PWD and B6 parental controls (*Figure 3*, *Figure 3—figure supplement 1*).

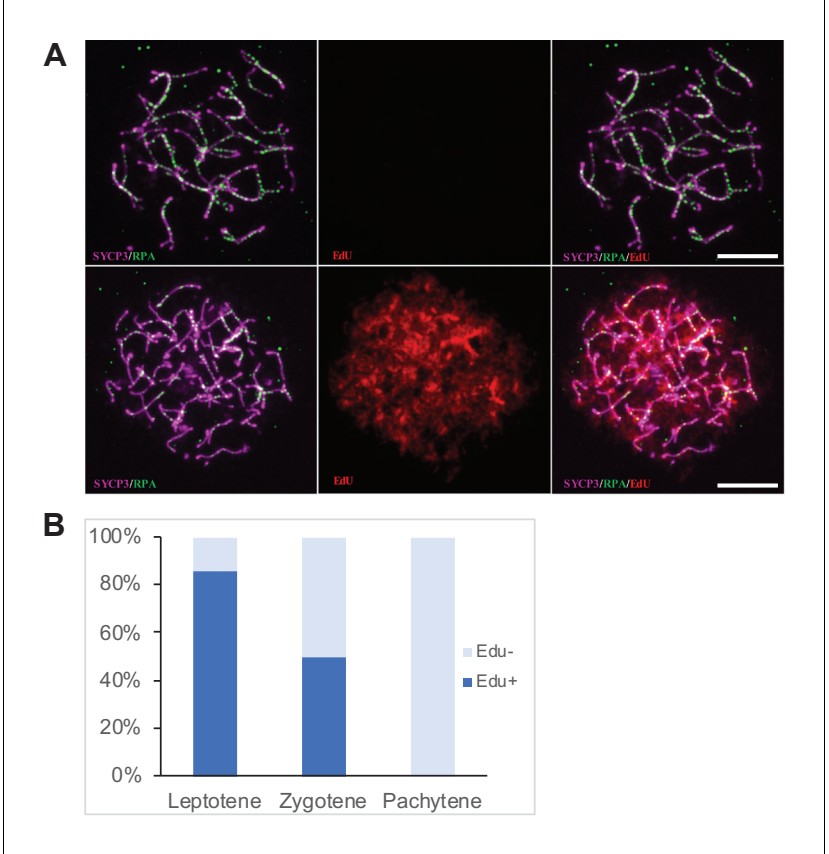

**Figure 2.** Determination of the cell cycle phase at the time of cisPt injection. (**A**) Forty h after EdU and CisPt injection, EdU-negative and positive zygonemas represent cells before and after the premeiotic S phase at the time of injection. Immunostaining of SYCP3 protein (violet) made chromosome axes visible. The RPA foci (green) associate with ssDNA of endogenous, SPO11-induced, and exogenous, cisPt-generated DSBs. Visualization of EdU-labeled DNA is based on the click reaction method (*Salic and Mitchison, 2008*). Scale bar 10 μM. (**B**) Proportion of EdU positive cells and EdU-negative cells at three prophase stages 40 hr after EdU treatment of eight males further analyzed in *Figures 3* and *4*. Numbers of examined cells: leptonemas 126, zygonemas 507, pachynemas 473.
DOI: https://doi.org/10.7554/eLife.42511.003
The following source data is available for figure 2:

**Source data 1.** Distribution of Edu + and EdU- spermatocytes at the first prophase 40 hr after EdU and cisPt injection.
DOI: https://doi.org/10.7554/eLife.42511.004

Unexpectedly, but in accord with *Moens et al. (2007)*, the RPA foci persisted in early pachynemas of fertile controls, but significantly dropped in mid pachynemas (median 38 and 14 RPA foci in PWD and B6 compared to 98 foci in PBF1, p<0.0001) and virtually disappeared at the late pachytene stage.

Surprisingly, when zygonemas were split to EdU-positive and negative, only EdU-positive cells showed a significant increase in the cisPt dosage-dependent RPA foci (*Figure 3B*, *Figure 3—source data 2*). The doses of 1 mg/kg, 5 mg/kg and 10 mg/kg of injected cisPt increased the median of RPA foci from 200.5 in controls to 239, 255 and 250, respectively (p=0.0043, 0.0026 and 0.0006, Mann-Whitney U test). As shown above, approximately half of the zygonemas were EdU-negative, apparently finishing S-phase before EdU and CisPt injection, while the EdU-positive zygonemas were in the mid or late S-phase at the time of the treatment. Since little is known about the timing of enzymatic removal of cisPt interstrand crosslinks (*Johnsson et al., 1995*), the formation of DSBs cannot

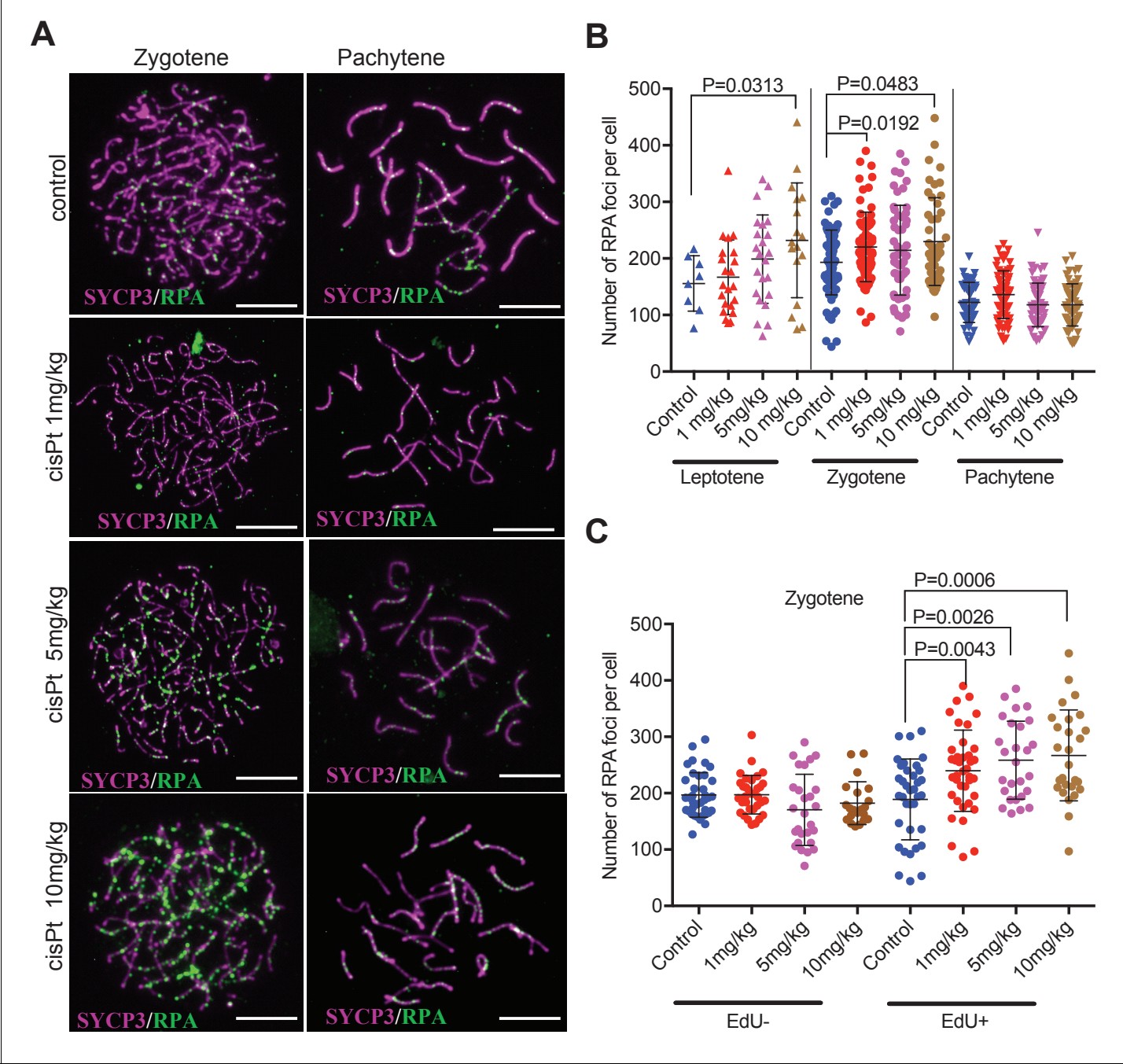

**Figure 3.** CisPt increases the frequency of exogenous DSBs monitored as RPA foci. (**A**) Images of RPA foci during zygotene and pachytene stages of the first meiotic prophase. RPA foci (green) harbored on chromosome axes visualized by immunostaining of SYCP3 protein (violet). Scale bar 10 μM. (**B**) Numbers of RPA foci per cell 40 hr after CisPt injection. In spite of the large variation of RPA foci between individual cells of the same cohort a significant increase (p<0.05) after cisPt application can be seen in leptotene and zygotene stages, while no indication of RPA foci increase is apparent at pachytene spermatocytes (**C**). When EdU-positive and -negative zygotene spermatocytes were analyzed separately, the enhancing effect of cisPt on the number of RPA foci was confined to EdU-positive cells. A significant dependence of RPA foci frequency on the dosage of cisPt is shown.

DOI: https://doi.org/10.7554/eLife.42511.005

The following source data and figure supplements are available for figure 3:

**Source data 1.** RPA foci in leptonemas (L), zygonemas (Z) and pachynemas (P) of PBF1 hybrid males treated with cisPt.

DOI: https://doi.org/10.7554/eLife.42511.008

**Source data 2.** RPA foci in EdU-negative (E-) and EdU-positive (E+) zygonemas of PBF1 hybrid males treated with cisPt.

DOI: https://doi.org/10.7554/eLife.42511.009

*Figure 3 continued on next page*

*Figure 3 continued*

**Figure supplement 1.** RPA foci in spermatocytes of PWD, B6 and (PWDxB6)F1 males.
DOI: https://doi.org/10.7554/eLife.42511.006
**Figure supplement 1—source data 1.** RPA foci in spermatocytes of PWD, B6 and PBF1 males.
DOI: https://doi.org/10.7554/eLife.42511.007

be precisely specified in respect of the end of DNA replication. The cisPt-induced DSBs could arise anytime during the meiotic S-phase and/or at the beginning of leptotene stage.

Since RPA is an ssDNA-binding protein, it could mark other forms of DNA damage beside DSBs (**Wang et al., 2005**); therefore, we quantified the foci of DNA meiotic recombination 1 (DMC1), a meiosis-specific strand exchange protein, which is recruited to SPO11-induced DSBs (**Figure 4A**). The DMC1 response to cisPt was similar to that of RPA. The combined EdU-positive and -negative zygonemas showed an enhancing effect of cisPt on the frequency of DMC1 foci (**Figure 4B**, **Figure 4—source data 1**). The median number of DMC1 foci increased from 215 to 241, 233 and 251 after 1, 5 and 10 mg/kg of cisPt (p=0.0208, 0.0263 and 0.0048, Mann-Whitney U), respectively. CisPt treatment did not influence the high frequency of DMC1 foci in PBF1 spermatocytes at pachytene stage. Contrary to RPA, the DMC1 foci significantly dropped (**Figure 4**, **Figure 4—figure supplement 1**) in early pachynemas of fertile PWD and B6 controls (median 60 and 31 DMC1 foci in PWD and B6 compared to 111 foci in PBF1, p<0.0001) and virtually disappeared at the mid pachytene stage (median 10 and 0 of DMC1 foci in PWD and B6 compared to 85 foci in PBF1, p<0.0001).

When split according to the EdU phenotype, the EdU-positive zygonemas showed a significant increase of DMC1 foci at all three concentrations (**Figure 4C**, **Figure 4—source data 2**), from 225.5 to 260.5, 269 and 259.5 foci, respectively (p=0.0025, 0.0192 and 0.0442, Mann-Whitney U test), while in EdU-negative cells the steady DMC1 increase became significant at 10 mg/kg dose (p=0.0246).

## CisPt treatment enhances meiotic synapsis of homologous chromosomes in sterile hybrids

Provided that the paucity of symmetric DSBs hotspots is indeed the main cause of meiotic synapsis failure (**Davies et al., 2016**; **Gregorova et al., 2018**) and that the increased frequency of DMC1 foci reflects cisPt-induced DSBs, then in spite of cytotoxicity of cisPt to proliferating cells, the exogenous DSBs should improve synapsis of the homologous chromosomes in the PBF1 testis. To verify this assumption, we analyzed the asynapsis rate by immunostaining the lateral elements of synaptonemal complexes with antibody against synaptonemal complex protein 3 (SYCP3) and unsynapsed axial cores of homologous chromosomes with antibodies specific for HORMA domain containing two proteins (HORMAD 2) (**Kogo et al., 2012**; **Wojtasz et al., 2012**) (**Figure 5A**). First, we tested in a pilot experiment the optimal effect of cisPt treatment by comparing the frequency pachynemas with a complete set of synapsed autosomes (hereafter 'fully synapsed pachynemas') at 4, 5, 7 and 8 days after a single injection of 10 mg/kg of cisPt. The percentage of pachynemas with fully synapsed bivalents dramatically increased from 5.66% in the control male (0 mg/kg of cisPt) to 48.78% and 46.70% in the males on the 7th and 8th day after treatment (**Figure 5B**, **Figure 5—source data 1**). In the next experiment, we combined the injection of cisPt (5 mg/kg or 10 mg/kg) with EdU (50 mg/kg) to distinguish the spermatogenic cells replicating their DNA at the moment of cisPt injection. For each cisPt dose, three males were sacrificed on day 8. The results confirmed the positive effect of cisPt on meiotic synapsis seen in the pilot experiment. The control males displayed the mean frequency of 8.61% (5.80; 12.12) (95% CI) of fully synapsed pachynemas in contrast to the males treated with 5 mg/kg and 10 mg/kg of cisPt, which showed a threefold increase of fully synapsed pachynemas, 24.68% (19.41; 30.49) (p=1.6 $\times$ 10$^{-9}$, Tukey's post-hoc test) and 28.71% (22.86; 35.08) (p=7.3 $\times$ 10$^{-13}$), respectively (**Figure 5C**, **Figure 5—source data 2**).

We assessed synapsis of homologous chromosomes at early, mid and late pachytene stages. However, for statistical evaluation the mid and late pachynemas were merged because of the scarcity of the latest stage. While the cisPt treatment caused a nonsignificant increase of fully synapsed early pachynemas, the enhancement of synapsis was dramatic in mid-late pachynemas (**Figure 5D**, **Figure 5—source data 2**). We assume that the efficiency of repair of cisPt-induced DSBs by

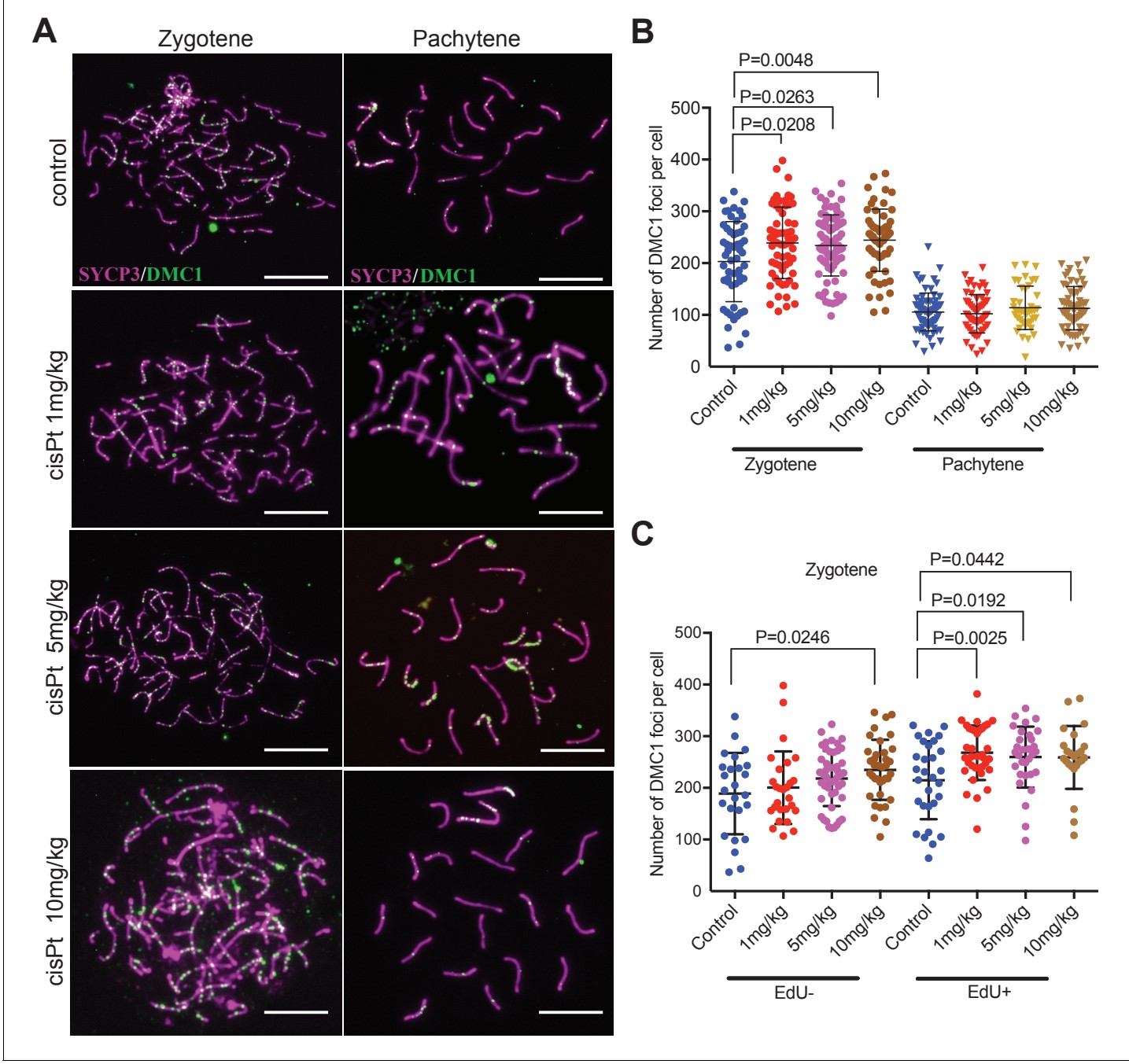

**Figure 4.** CisPt increases the frequency of exogenous DSBs monitored as DMC1 foci. (**A**) Images of DMC1 foci (green) during zygotene and pachytene stages of the first meiotic prophase. Scale bar 10 μM. (**B**) The numbers of DMC1 foci 40 hr after CisPt injection increase in a dose-dependent manner in zygonemas but do not change in spermatocytes at the pachytene stage. (**C**) The enhancing effect of cisPt on the number of DMC1 foci at the zygotene stage is barely significant in EdU-negative cells but detectable at all three cisPt doses in EdU-positive zygonemas.
DOI: https://doi.org/10.7554/eLife.42511.010

The following source data and figure supplements are available for figure 4:

**Source data 1.** DMC1 foci in zygonemas (Z) and pachynemas (P) of PBF1 hybrid males treated with cisPt.
DOI: https://doi.org/10.7554/eLife.42511.013

**Source data 2.** DMC1 foci in EdU-negative (E-) and EdU-positive (E+) zygonemas of PBF1 hybrid males treated with cisPt.
DOI: https://doi.org/10.7554/eLife.42511.014

**Figure supplement 1.** DMC1 foci in spermatocytes of PWD, B6 and (PWDxB6)F1 males.
DOI: https://doi.org/10.7554/eLife.42511.011

*Figure 4 continued on next page*

*Figure 4 continued*

**Figure supplement 1—source data 1.** DMC1 foci in spermatocytes of PWD, B6 and PBF1 males.
DOI: https://doi.org/10.7554/eLife.42511.012

standard homologous recombination is low in general and/or that a significant fraction of spermatocytes carrying them do not survive until early pachytene stage. Those spermatocytes with multiple asynapsed autosomes that survive to the early pachytene stage are mostly eliminated before reaching the mid-late pachytene stage as reported previously (*Bhattacharyya et al., 2013*). This multiple filtering effect thus enhances the apparent efficiency of cisPt monitored as a proportion of fully synapsed pachynemas at the mid-late stage.

When divided according the cell cycle stage at the time of cisPt injection, the synapsis was marginally significantly more frequent (p=0.0433, GLM model) in EdU-negative than in EdU-positive pachynemas (*Figure 5E*), on average 1.69 times (1.02; 2.84, 95 % CI; *Figure 5—source data 2*). The EdU-negative fully synapsed pachynemas could arise from a subset of cells with exogenous DSBs generated at leptotene/early zygotene at the time of cisPt and EdU injection. It is tempting to suggest that also the EdU-positive cells that were at the preleptotene S-phase at the time of injection removed the cisPt induced ICLs at early leptotene stage.

No sperm was found in the ductus epididymis of the males 30 days after injection. Histological sections showed atrophy of seminiferous tubules caused by the lethal effect of cisPt on proliferating spermatogonia and somatic cells of seminiferous tubules (not shown). Beside the increased frequency of fully synapsed pachynemas, a short-term effect was apparent from the increased relative incidence of late pachynemas. While in untreated control hybrids the late pachynemas represented 2.03% (4/107) of all pachynemas recorded from the meiotic spreads, the frequency increased to 12.12% (28/231) and 13.10% (30/229) after 5 mg/kg and 10 mg/kg cisPt treatment, respectively.

## Improved synapsis of meiotic chromosomes by exogenous DNA DSBs points to the insufficient number of properly repaired DSBs as the ultimate cause of meiotic asynapsis and hybrid sterility

The genetic network controlling incomplete synapsis of homologous chromosomes, early meiotic arrest, and male sterility of mouse inter-subspecific PBF1 hybrids is formed by three components, *Prdm9PWD/B6* heterozygosity (*Mihola et al., 2009*), PWD allele at the *Hstx2* locus on Chromosome X (*Storchová et al., 2004*; *Bhattacharyya et al., 2014* for review, see *Forejt et al., 2012*), and autosomal PWD/B6 heterozygosity (*Dzur-Gejdosova et al., 2012*; *Gregorova et al., 2018*). While the molecular mechanism of the *Hstx2* action is still unclear, four mutually nonexclusive explanations of PRDM9-controlled meiotic arrest have been proposed. Originally, we hypothesized that a divergence of fast evolving noncoding DNA and/or RNA sequences could interfere with the homology search of single-strand 3' ends on a heterosubspecific template during the DSB repair, thus interfering with chromosome synapsis (*Bhattacharyya et al., 2014*). However, our hypothesis offered no explanation for the role of *Prdm9* in the presumed impairment of homology search. Later, using our PBF1 hybrid sterility model, *Davies et al. (2016)* found that ~70% of PRDM9-directed hotspots were enriched on a 'nonself' chromosome (e.g. PRDM9B6 on PWD chromosome and vice versa). DSBs in these hotspots are difficult to repair or they repair too late, perhaps using sister chromatids as a template (*Faieta et al., 2016*; *Li et al., 2018*). Chromosomal distribution of asymmetric DSB hotspots correlated well with the asynapsis rate of particular chromosomes (*Davies et al., 2016*; *Gregorova et al., 2018*) and indicated that the insufficient number of DSBs generated at symmetric hotspots may limit their pairing and normal progression of spermatogenesis. The present results show that, indeed, addition of repairable, non-DSBs in the form of exogenous DSBs significantly improved the faulty synapsis of homologous chromosomes.

Another possible mechanism explaining the role of *Prdm9* in meiotic arrest points to a significant enrichment of the default, PRDM9-independent DSB hotspots in PBF1 spermatocytes (*Smagulova et al., 2016*). Such hotspots were observed in *Prdm9-/-* sterile males and are preferentially located in promoters and other regulatory sequences. This observation could indicate functional deficiency of PRDM9 in hybrid males, such as inefficient PRDM9 multimers (*Baker et al., 2015*; *Altemose et al., 2017*) the improvement of which is difficult to envisage by adding

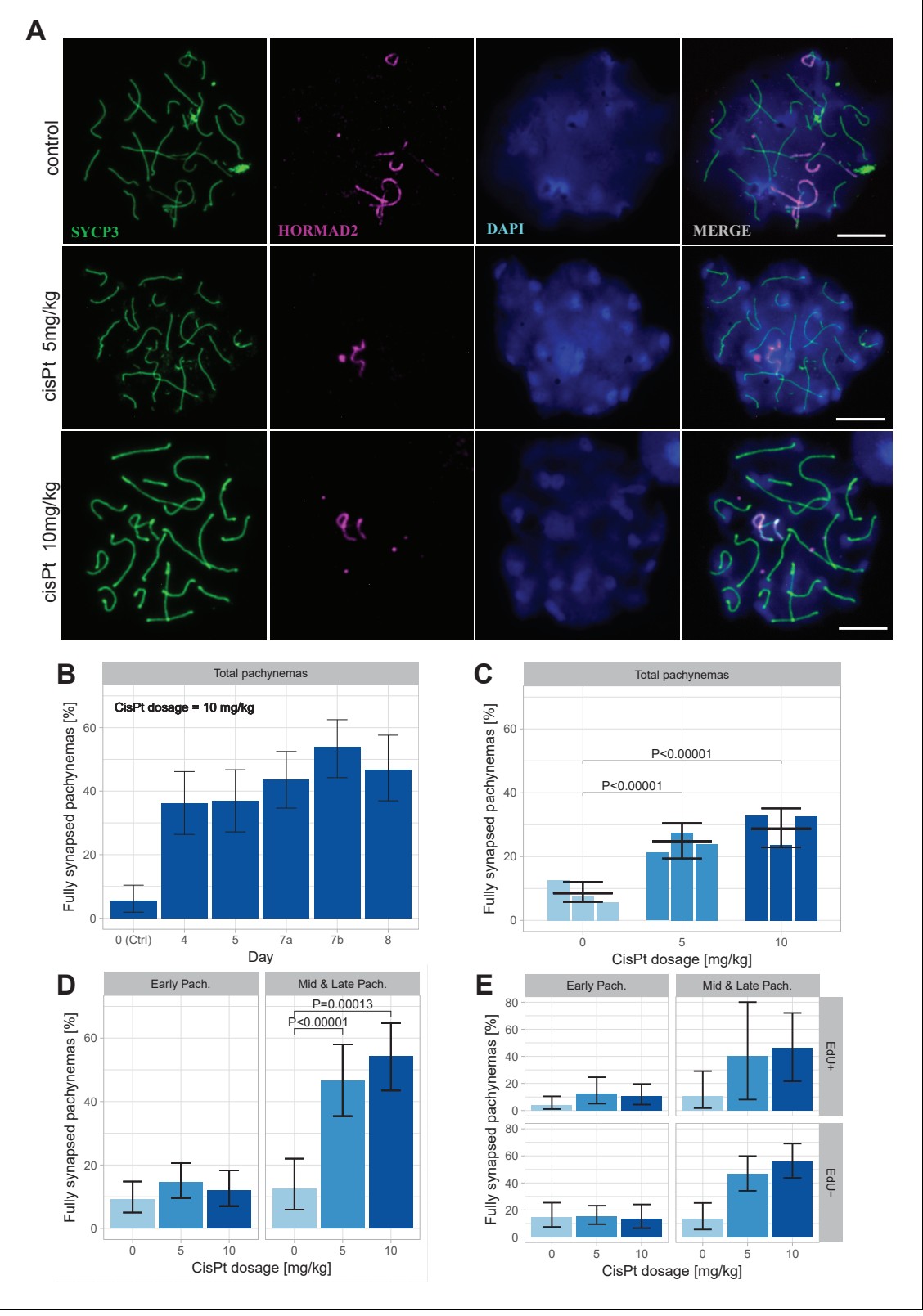

**Figure 5.** CisPt supports full synapsis of homologous chromosomes at the pachytene stage. (**A**) Examples of control and cisPt-treated pachynemas 8 days after cisPt injection. Unsynapsed parts of X and Y chromosomes (5 and 10 mg cisPt/kg) together with unsynapsed autosomal axes (control) were visualized by anti-HORMAD2 antibody (violet). Axial elements of unsynapsed chromosomes and lateral elements of synaptonemal complexes were decorated by anti-SYCP3 antibody (green) and DNA painted by DAPI. The displayed spermatocytes are at early (control and 5 mg/kg) and late (10 mg/

*Figure 5 continued on next page*

*Figure 5 continued*

kg) pachytene stage. Scale bar 10 µM. (B) Frequency of fully synapsed pachynemas +-S.E. (based on GLMM model), after a single dose of 10 mg/kg of cis Pt; a pilot experiment. Treated males were sacrificed from day 4 to day eight after injection. Each column represents a single male. (C) CisPt dosage-dependent improvement of meiotic chromosome synapsis. Eight days after cisPt injection the percentage of fully synapsed pachynemas significantly increased after cisPt treatment (based on GLMM model and Tukey's post-hoc test). (D) The effect of cisPt on meiotic synapsis is apparent in mid and late pachytene stages. (E) The meiotic synapsis is slightly enhanced in EdU-negative pachynemas. See text for details.

DOI: https://doi.org/10.7554/eLife.42511.015

The following source data is available for figure 5:

**Source data 1.** The effect of cisPt (10 mg/kg) on chromosome synapsis at pachytene in sterile PBF1 hybrid males 0 to 8 days after treatment.
DOI: https://doi.org/10.7554/eLife.42511.016

**Source data 2.** Evaluation of meiotic chromosome synapsis at pachytene stage in PBF1 males treated with cisPt.
DOI: https://doi.org/10.7554/eLife.42511.017

exogenous DSBs. Finally, since a recent report uncovered about 30% of PRDM9-controlled DSBs in repetitive sequences including transposons, their homology at nonallelic sites could destabilize genome integrity and interfere with the DSB repair (*Yamada et al., 2017*). Such a mechanism could operate independently of and in parallel with the symmetric DSB-dependent pachytene checkpoint.

To conclude, our results complement our previous findings (*Gregorova et al., 2018*) bringing new evidence for the deficiency in properly repaired DSBs as one of the major causes of meiotic asynapsis and male sterility of PBF1 inter-subspecific hybrids. Although our results do not exclude the role of PRDM9 default or retroposon-directed hotspots in meiotic failure, they bring a new and independent evidence in favor of DSB hotspot asymmetry caused by PRDM9 hotspot erasure as the main cause of chromosome asynapsis and meiotic arrest in PBF1 intersubspecific hybrid sterility.

# Materials and methods

**Key resources table**

| Reagent type (species) or resource | Designation | Source or reference | Identifiers | Additional information |
|---|---|---|---|---|
| Strain, strain background (*Mus m. domesticus*) | C57BL/6J | The Jackson Laboratory | Stock No: 000664 \| Black 6 | Laboratory inbred strain, predominantly of *Mus m. domesticus* origin |
| Strain, strain background (*Mus m. musculus*) | PWD/Ph | Institute of Molecular Genetics, ASCR, Prague | N/A | Wild-derived inbred strain of *Mus m. musculus* origin |
| Antibody | anti SYCP3 (mouse monoclonal) | Santa Cruz Biotechnology | Santa Cruz: sc-74569; RRID:AB_2197353 | (1:50) |
| Antibody | anti HORMAD2 (rabbit polyclonal) | gift from Attila Toth | N/A | (1:700) |
| Antibody | anti HORMAD2 (rabbit polyclonal , C-18) | Santa Cruz Biotechnology | Santa Cruz:sc-82192; RRID:AB_2121124 | (1:500) |
| Antibody | Anti RPA (rabbit polyclonal) | gift from Willy M. Baarends | N/A | (1:150) |
| Antibody | Anti DMC1 ((rabbit polyclonal) | Santa Cruz | Santa Cruz: SC-22768; RRID:AB_2277191 | (1:300) |
| Antibody | anti-rabbit IgG - AlexaFluor568 (goat polyclonal) | Molecular Probes | Molecular Probes: A-11036; RRID:AB_10563566 | (1:500) |
| Antibody | anti-mouse IgG - AlexaFluor647 (goat polyclonal) | Molecular Probes | Molecular Probes: A-21235; RRID:AB_141693 | (1:500) |

*Continued on next page*

*Continued*

| Reagent type (species) or resource | Designation | Source or reference | Identifiers | Additional information |
|---|---|---|---|---|
| Other | normal goat serum from healthy animals | Chemicon | Chemicon: S26-100ML | |
| Commercial assay or kit | Base-click EdU IV Imaging kit 555S | Baseclick | BaseClick: BCK-EdU555 | |
| Chemical compound, drug | cisplatin | Sigma-Aldrich-Merck | Sigma-Aldrich: C2210000 | 1, 5, or 10 mg/kg |

## Mice, cisplatin and EdU application

The mice were maintained at the Institute of Molecular Genetics in Prague and Vestec, Czech Republic. The project was approved by the Animal Care and Use Committee of the Institute of Molecular Genetics AS CR, protocol No 141/2012. The principles of laboratory animal care, Czech Act No. 246/1992 Sb., compatible with EU Council Directive 86/609/EEC and Appendix of the Council of Europe Convention ETS, were observed. The origin of the PWD/Ph and C57BL/6J mouse strains, the PBF1 hybrids and their handling were described in the previous paper (*Gregorova et al., 2018*). Cisplatin (Merck, C2210000) was freshly dissolved in 0.9% NaCl, 1 mg/ml, and intraperitoneally injected at 1, 5 or 10 mg per 1 kg of body weight. EdU was dissolved in PBS and injected at 50 mg/kg.

## Immunostaining and image capture

For immunocytochemistry, the spread nuclei were prepared as described (*Anderson et al., 1999*) with modifications. Briefly, single-cell suspension of spermatogenic cells in 0.1M sucrose with protease inhibitors (Roche) was dropped on 1% paraformaldehyde-treated slides and allowed to settle for 3 hr in a humidified box at 4°C. After brief $H_2O$ and PBS washing and blocking with 5% goat sera in PBS (vol/vol), the cells were immunolabeled using a standard protocol with the following antibodies: anti-HORMAD2 (1:700, rabbit polyclonal antibody, gift from Attila Toth) and SYCP3 (1:50, mouse monoclonal antibody, Santa Cruz, #74569). Secondary antibodies were used at 1:500 dilutions and incubated at room temperature for 60 min; goat anti-rabbit IgG-AlexaFluor568 (MolecularProbes, A-11036) and goat anti-mouse IgG-AlexaFluor647 (MolecularProbes, A-21235). Visualization of EdU-labeled nuclei was done using an EdU in vivo kit (Baseclick) according to the manufacturer's instructions. The images were acquired and examined in a Nikon Eclipse 400 microscope with a motorized stage control using a Plan Fluor objective, 60x (MRH00601; Nikon) and captured using a DS-QiMc monochrome CCD camera (Nikon) and the NIS-Elements program (Nikon). To quantify RPA and DMC1 foci in spread nuclei we used ImageJ (Wayne Rasband, National Institute of Health, USA, http://imagej.nih.gov/ij). Images were processed using the Adobe Photoshop CS software (Adobe Systems). The estimates of the mean asynapsis rate, their standard errors and 95% confidence intervals were based on the Generalized Linear Mixed Model (GLMM) described in the previous paper (*Gregorova et al., 2018*).

## Acknowledgements

We thank Vladana Fotopulosova and Diana Lustyk for the help with meiotic analyses, Petr Jansa and Vaclav Gergelits and Emil Parvanov for helpful comments and help in preparation of the figures. Statistical evaluation of the synapsis data by Generalized Linear Mixed Model was kindly done by Vaclav Gergelits. This work was supported by Czech Science Foundation grant 16–01969S and by the LQ1604 project of NSPII from the Ministry of Education, Youth and Sports of the Czech Republic.

## Additional information

### Funding

| Funder | Grant reference number | Author |
| --- | --- | --- |
| Charles University Grant Agency | 17115 | Barbora Valiskova |
| Grantová Agentura České Republiky | 16-01969S | Jiri Forejt |
| Ministry of Education, Youth and Sports | LQ1604 project of NSPII | Jiri Forejt |

The funders had no role in study design, data collection and interpretation, or the decision to submit the work for publication.

### Author contributions

Liu Wang, Formal analysis, Investigation, Methodology, Writing—original draft, Writing—review and editing; Barbora Valiskova, Validation, Investigation, Visualization, Methodology, Writing—original draft; Jiri Forejt, Conceptualization, Formal analysis, Funding acquisition, Investigation, Writing—original draft, Project administration, Writing—review and editing

### Author ORCIDs

Jiri Forejt (iD) http://orcid.org/0000-0002-2793-3623

### Ethics

Animal experimentation: The project was approved by the Animal Care and Use Committee of the Institute of Molecular Genetics AS CR, protocol No 141/2012. The principles of laboratory animal care, Czech Act No. 246/1992 Sb., compatible with EU Council Directive 86/609/EEC and Appendix of the Council of Europe Convention ETS, were observed.

### Decision letter and Author response

Decision letter https://doi.org/10.7554/eLife.42511.021
Author response https://doi.org/10.7554/eLife.42511.022

## Additional files

### Supplementary files

• Transparent reporting form
DOI: https://doi.org/10.7554/eLife.42511.018

### Data availability

All data generated or analyzed during this study are included in the manuscript and source files.

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
