## [Decision Letter]

Thank you for submitting your article "Cisplatin-induced DNA double-strand breaks promote meiotic chromosome synapsis in PRDM9-controlled hybrid sterility" for consideration by *eLife*. Your article has been reviewed by two peer reviewers, and the evaluation has been overseen by Patricia Wittkopp as the Senior and Reviewing Editor. The reviewers have opted to remain anonymous.

The reviewers have discussed the reviews with one another and the Reviewing Editor has drafted this decision to help you prepare a revised submission.

Summary:

This "Research Advance" report by Forejt and colleagues provides orthogonal support for the hypothesis that male hybrid sterility with allelic incompatibility between Prdm9 and chromosomal haplotypes is caused by insufficient "asymmetric" DSBs, leading to failed homologous chromosome synapsis that triggers the meiotic checkpoint. This short manuscript is in support of a previous *eLife* paper that had remarkable genetic support for this model.

The basic idea is that both homologs need to receive about equal numbers of DSBs to promote interhomolog repair. The argument goes that if one homolog gets the majority of DSBs (because the PRDM9 allele's binding sites are underrepresented on one homolog), these will tend to be repaired by intersister recombination, which isn't productive in driving synapsis. It isn't clear to me why this would be, and the paper doesn't really address this potential mechanism. Nevertheless, the authors show that exogenous DSBs induced by cisplatin, presumably which occur randomly across all chromosome homologs and thus are "symmetric," substantially rescues synapsis and spermatocytes.

The manuscript itself is nicely done, and the results are convincing that cisplatin-induced DSBs rescue synapsis in a substantial fraction of hybrid spermatocytes. This would be an interesting addition to the meiosis literature, in that it supports the findings of older papers indicating that exogenous DSBs can, to some extent, substitute for SPO11 DSBs. Regarding its contribution to the hybrid sterility literature, these results strongly support the idea that a DSB deficiency underlies the synapsis defect. However, the manuscript does not provide direct evidence for the "asymmetric DSB" hypothesis; this would require a comparison of intersister vs. interhomolog repair of DSBs either genome-wide or in chromosomal intervals that are particularly susceptible to asynapsis in hybrids. This is an important shortfall of the paper with respect to linking to the "parent" paper, but it isn't clear how this can actually be addressed directly.

Essential revisions:

1) Add a cogent description (or cartoon figure) of the symmetric DSB model. This concept is confusing and not widely known in the meiosis field. From reading this in isolation, one might erroneously assume that symmetric DSBs are those that occur on both homologs at the same hotspot in the same spermatocyte, whereas asymmetric ones only occur on one homolog. Another interpretation is that symmetric sites are where PRDM9 binds the same locus of both homologs, but only one receives a DSB.

2) Propose an explanation for why asymmetric DSBs are primarily repaired via sister chromatid recombination…this implies that such DSBs along chromosome intervals somehow communicate with one another, i.e., that if all/most DSBs are on one homolog, then they are repaired via the sisters, but if the DSBs are distributed equally between homologs, then interhomolog repair is favored. Are there hints from other model organisms in which intersister events can be monitored directly?

3) Address the following concerns related to the data in Figure 4:i) the conclusion that cisplatin improves synapsis at "mid-late" pachynema requires accurate substaging of cells, which is not the case based on Figure 4A. The bottom cell is clearly at late pachynema, as shown by the bulging chromosome ends, behaviour of the XY bivalent, and DAPI-staining. But the top and middle cells are at early pachynema (compare the chromosome morphology and DAPI). The issue is magnified by the apparently very high levels of asynapsis in the untreated mice at mid-late pachynema (Figure 4E). As the authors and others have found, cells with asynapsis are eliminated at mid pachynema, so I'm not sure how these high levels of asynapsis can be possible at late prophase I. A systematic problem in stage-matching should be considered here.

ii) It isn't clear to me why the authors don't also find improved synapsis at early pachynema. Sure, the frequency of asynapsis will be higher at early than at mid-late pachynema because of the midpachytene checkpoint, but shouldn't a difference be observed between the cisplatin-treated and -untreated mice at early pachynema?

4) The effect of cisplatin on other aspects of the hybrid sterile phenotype are not presented. The authors state in data not shown that cisplatin causes atrophy due to toxic effect. But couldn't they assay in the short-term whether the number of cells reaching late pachynema after treatment is increased? And how do toxic effects of cisplatin on pachytene substages influence the interpretations in Figure 4?

5) EdU treatment.

i) To the non-initiated, the expectations and interpretations of the EdU+ and – cells will not be easy to understand. The authors could do a better job of explaining this.

ii) Why are DSB counts assayed at zygonema and pachynema but not leptonema? Isn't the latter stage the important one to focus on given the DSB hypothesis?

6) Clarify the following: Why is there so much asynapsis in the early pachytenes (assuming correct staging), even with CP treatment? Is it possible that the timing was such that this cohort was derived from cells that weren't in S phase? And so many asynapsed "Mid-late" pachytenes doesn't make sense in terms of checkpoint elimination…perhaps H1t staining was needed? These potential issues with stage-matching and the lack of an objective measure (beyond incidence of full synapsis) with which to ascertain that cisplatin rescues F1 hybrid cells are major concerns.

---

## [Author Response]

[…] The basic idea is that both homologs need to receive about equal numbers of DSBs to promote interhomolog repair.

We are sorry that we have not been clear enough when describing the asymmetric hotspot hypothesis. An unequal mean number of DSBs on homologs is not the explanation of our model of hybrid sterility. The proposed mechanism of asymmetric DSB hotspots (Davies et al., 2016) is based on the subspecies-specific erasure of PRDM9 binding sites, as demonstrated in the scheme newly added in Figure 1. The asymmetry in sterile F1 hybrids results in unrepairable or hardly repairable DSBs trying to use the homologous chromosome as a template. These unrepaired DSBs and the failure of pairing due to the deficiency of non-sister repaired DSBs results in elimination of meiotic cells.

The argument goes that if one homolog gets the majority of DSBs (because the PRDM9 allele's binding sites are underrepresented on one homolog), these will tend to be repaired by intersister recombination, which isn't productive in driving synapsis. It isn't clear to me why this would be, and the paper doesn't really address this potential mechanism. Nevertheless, the authors show that exogenous DSBs induced by cisplatin, presumably which occur randomly across all chromosome homologs and thus are "symmetric," substantially rescues synapsis and spermatocytes.

We believe that this premise is based on misunderstanding the asymmetric hotspot hypothesis. The *Prdm9* allele-specific DSBs occuring only on one homolog (being heterozygous according to Grey, Baudat and de Massy, 2018) are present on *both* homologs, PWD on B6 chromosome and B6 on PWD chromosome (see Figure 2A in Davies et al., 2016). The asymmetry model is based on the subspecies-specific erosion of PRDM9 binding sites due to the meiotic drive (see the newly added Figure 1). If the majority of DSBs cannot be repaired by recombination (CO or NCO) from the homologous chromosomes and the number of sucessfully repaired DSBs falls below a certain threshold, then the 100% efficiency of homologous chromosome synapsis becomes compromised.

The manuscript itself is nicely done, and the results are convincing that cisplatin-induced DSBs rescue synapsis in a substantial fraction of hybrid spermatocytes. This would be an interesting addition to the meiosis literature, in that it supports the findings of older papers indicating that exogenous DSBs can, to some extent, substitute for SPO11 DSBs. Regarding its contribution to the hybrid sterility literature, these results strongly support the idea that a DSB deficiency underlies the synapsis defect. However, the manuscript does not provide direct evidence for the "asymmetric DSB" hypothesis; this would require a comparison of intersister vs. interhomolog repair of DSBs either genome-wide or in chromosomal intervals that are particularly susceptible to asynapsis in hybrids. This is an important shortfall of the paper with respect to linking to the "parent" paper, but it isn't clear how this can actually be addressed directly.

We think that the argument is based on misunderstanding the asymmetric hotspot concept. The question we asked was whether, as predicted by the asymmetry hypothesis, exogenous DSBs can improve synapsis of homologous chromosomes in sterile interspecies hybrids by increasing the number of presumably repairable DSBs. The apoptosis of primary spermatocytes can be provoked by persisting unrepaired DSBs, or to be a response to the failure of meiotic synapsis. We newly compared the rates of DMC1 and RPA foci at the pachytene stage in the fertile untreated parents, B6 and PWD, with sterile (PWD x B6)F1 hybrids (Figure 3—figure supplement 1 and Figure 4—figure supplement 1) to emphasize the large numbers of persisting DSBs in pachynemas of sterile F1 hybrids. The results are summarized in the subsection “CisPt induced DSBs in early meiotic prophase of sterile male hybrids”, second and fourth paragraphs. To our best knowledge, no method is available to provide direct evidence for intersister chromatid repair in mouse/mammalian meiosis.

Essential revisions:1) Add a cogent description (or cartoon figure) of the symmetric DSB model. This concept is confusing and not widely known in the meiosis field. From reading this in isolation, one might erroneously assume that symmetric DSBs are those that occur on both homologs at the same hotspot in the same spermatocyte, whereas asymmetric ones only occur on one homolog. Another interpretation is that symmetric sites are where PRDM9 binds the same locus of both homologs, but only one receives a DSB.

We thank the reviewers for the comment. We agree that a cogent description of the symmetric DSB model was missing, which might have led to misunderstandings. We added a schematic drawing of a pair of homologous chromosomes with the asymmetric DSBs in the new Figure 1 and supplied the detailed legend.

2) Propose an explanation for why asymmetric DSBs are primarily repaired via sister chromatid recombination…this implies that such DSBs along chromosome intervals somehow communicate with one another, i.e., that if all/most DSBs are on one homolog, then they are repaired via the sisters, but if the DSBs are distributed equally between homologs, then interhomolog repair is favored. Are there hints from other model organisms in which intersister events can be monitored directly?

In the manuscript we do not propose that the asymmetric DSBs are primarily repaired via sister chromatid recombination. Rather, we think that the asymmetric DSBs primarily remain unrepaired, as mentioned above. Our only reference to sister chromatid recombination is: “DSBs in these hotspots are difficult to repair or they repair too late, perhaps using sister chromatids as a template (Faieta et al. 2016; Li et al. 2018).” However, we do present direct cytological evidence for the persisting DSBs at the pachytene stage and for asynapsis, both of them being able to arrest the first meiotic division on its own. Contrary to the clearance of DSBs on unpaired parts of X and Y chromosomes or on certain chromosomal translocations where DMC1/RAD51 foci disappear by late pachytene most likely by inter-sister-chromatid repair, the high numbers of RPA and DMC1 foci persist through the pachytene stage in (PWD x B6)F1 hybrid males. The cells are eliminated before the DSBs could have been repaired by inter-sister recombination (see newly added Figure 3—figure supplement 1 and Figure 4—figure supplement 1). It is probable that the sister chromatid recombination also contributes to the reduction of DSB numbers at the pachytene stage (compared with zygonemas), but we do not possess tools to experimentally verify this idea.

3) Address the following concerns related to the data in Figure 4:i) the conclusion that cisplatin improves synapsis at "mid-late" pachynema requires accurate substaging of cells, which is not the case based on Figure 4A.

The conclusion is based on the quantitative evaluation of 645 pachynemas classified as early, middle and late pachynemas (See Figure 5—source data 2. Figure 4 became Figure 5 after revision). Figure 5A illustrates the reliability of the method to detect asynapsis but legend does not specify the pachytene stage.

The bottom cell is clearly at late pachynema, as shown by the bulging chromosome ends, behaviour of the XY bivalent, and DAPI-staining. But the top and middle cells are at early pachynema (compare the chromosome morphology and DAPI).

We agree with the reviewer that the cell is a late pachynema and admit the pachytene stages were not specified in the legend to Figure 5A. To avoid misunderstanding, the pachytene substage of all three cells is now added to the legend.

The issue is magnified by the apparently very high levels of asynapsis in the untreated mice at mid-late pachynema (Figure 4E). As the authors and others have found, cells with asynapsis are eliminated at mid pachynema, so I'm not sure how these high levels of asynapsis can be possible at late prophase I. A systematic problem in stage-matching should be considered here.

Previously, we analysed the ratio of asynaptic cells in PBF1 hybrids (Bhattacharyya et al., 2013 Figure 2C). It shows that the cells with four or less univalents can reach mid pachytene and a few of them late pachytene. Pachynemas with > 4 univalents were detected at early pachytene but were missing later. In this manuscript, we deliberately classified pachynemas into two groups, fully synapsed and pachynemas with asynapsis, to quantify the overall effect of cisPt on meiotic synapsis.

ii) It isn't clear to me why the authors don't also find improved synapsis at early pachynema. Sure, the frequency of asynapsis will be higher at early than at mid-late pachynema because of the midpachytene checkpoint, but shouldn't a difference be observed between the cisplatin-treated and -untreated mice at early pachynema?

We thank the reviewer for the comment. Indeed, we did not expect such a difference between early and mid-late pachynemas and failed to comment on it in the text. The detailed explanation was added to the second paragraph of the subsection “CisPt treatment enhances meiotic synapsis of homologous chromosomes in sterile hybrids”. See also our response to point 6.

4) The effect of cisplatin on other aspects of the hybrid sterile phenotype are not presented. The authors state in data not shown that cisplatin causes atrophy due to toxic effect. But couldn't they assay in the short-term whether the number of cells reaching late pachynema after treatment is increased? And how do toxic effects of cisplatin on pachytene substages influence the interpretations in Figure 4?

We thank the reviewers for the suggestion to focus on the late pachynemas after treatment. The comparison indeed shows the increase of late pachynemas. The analysis was included as follows “Beside the increased frequency of fully synapsed pachynemas, a short-term effect was apparent from the increased relative incidence of late pachynemas. While in untreated control hybrids the late pachynemas represented 2.03% (4/107) of all pachynemas recorded from the meiotic spreads, the frequency increased to 12.12% (28/231) and 13.10% (30/229) after 5 mg/kg and 10 mg/kg cisPt treatment, respectively”.For the interpretation of the toxic effect of cisPt in Figure 5, see also our answer to Essential revisions No 6.

5) EdU treatment.i) To the non-initiated, the expectations and interpretations of the EdU+ and – cells will not be easy to understand. The authors could do a better job of explaining this.

We thank the reviewer for comment. The rationale of EdU treatment was explained as follows: “we treated the adult (4-8 weeks) PBF1 hybrid males with cisPt and with 5-ethynyl-2’-deoxyuridine (EdU), a nucleoside analog of thymidine (Salic and Mitchison, 2008) to distinguish the spermatogenic cells replicating their DNA at the moment of cisPt injection.”

We also added an additional sentence of explanation to the first paragraph of the subsection “CisPt treatment enhances meiotic synapsis of homologous chromosomes in sterile hybrids”.

ii) Why are DSB counts assayed at zygonema and pachynema but not leptonema? Isn't the latter stage the important one to focus on given the DSB hypothesis?

We had two reasons for assaying zygonemas. The spread preparations contain very few spermatocytes at the leptotene stage, and DMC1 and RAD51 foci are more numerous at zygotene than in leptotene (see e.g. Cole et al., Nat Cell Biol. 2012 or Kauppi et al., Gen Dev 2013). The most probable reason is that the foci accumulate gradually during the leptotene stage and survive for the major part of the zygotene stage.

6) Clarify the following: Why is there so much asynapsis in the early pachytenes (assuming correct staging), even with CP treatment? Is it possible that the timing was such that this cohort was derived from cells that weren't in S phase?

We thank the reviewers for the comment. The missing explanation was added on as follows: “We assume that the efficiency of repair of cisPt-induced DSBs by meiotic homologous recombination is low and/or that a significant fraction of spermatocytes carrying them do not survive until early pachytene stage. […] This multiple filtering effect thus enhances the apparent efficiency of cisPt monitored as a proportion of fully synapsed pachynemas at mid-late stage.”

And so many asynapsed "Mid-late" pachytenes doesn't make sense in terms of checkpoint elimination…perhaps H1t staining was needed?

The question was answered above: The ratio of asynaptic cells in PBF1 hybrids is given in Bhattacharyya (Bhattacharyya et al., 2013 Figure 2C). It shows that only cells with four or less univalents can reach mid pachytene and a few of them late pachytene. Pachynemas with > 4 univalents were detected only in the early pachytene stage. In this manuscript, we deliberately classified pachynemas as fully synapsed and pachynemas with asynapsis to verify the overall effect of cisPt on meiotic synapsis.

These potential issues with stage-matching and the lack of an objective measure (beyond incidence of full synapsis) with which to ascertain that cisplatin rescues F1 hybrid cells are major concerns.

We hope the concerns about the stage-matching were dispelled. We checked the effect of exogenous DSBs at the level of synapsis, which was verified. The ratio of the three pachytene substages was added to show the significant increase of late pachynemas after cisPt treatment. See also the answer to point 4.